# Macrophage-Specific, *Mafb*-Deficient Mice Showed Delayed Skin Wound Healing

**DOI:** 10.3390/ijms23169346

**Published:** 2022-08-19

**Authors:** Yuri Inoue, Ching-Wei Liao, Yuki Tsunakawa, I-Lin Tsai, Satoru Takahashi, Michito Hamada

**Affiliations:** 1Department of Anatomy and Embryology, Faculty of Medicine, University of Tsukuba, 1-1-1 Tennodai, Tsukuba 305-8575, Ibaraki, Japan; 2Doctoral Program in Biomedical Sciences, Graduate School of Comprehensive Human Sciences, University of Tsukuba, 1-1-1 Tennodai, Tsukuba 305-8575, Ibaraki, Japan; 3Ph.D. Program in Human Biology, School of Integrative and Global Majors, University of Tsukuba, 1-1-1 Tennodai, Tsukuba 305-8575, Ibaraki, Japan; 4Global Innovation Joint-Degree Program, International Joint Degree Master’s Program, Agro-Biomedical Science in Food and Health, College of Medicine, National Taiwan University (NTU GIP-TRIAD), No. 1, Sec. 4, Roosevelt Rd., Taipei 10617, Taiwan; 5International Institute for Integrative Sleep Medicine (WPI-IIIS), University of Tsukuba, 1-1-1 Tennodai, Tsukuba 305-8575, Ibaraki, Japan

**Keywords:** macrophage, MAFB, wound healing, skin, Arg1, CCL12, CCL2

## Abstract

Macrophages play essential roles throughout the wound repair process. Nevertheless, mechanisms regulating the process are poorly understood. MAFB is specifically expressed in the macrophages in hematopoietic tissue and is vital to homeostatic function. Comparison of the skin wound repair rates in macrophage-specific, MAFB-deficient mice (*Mafb^f/f^*::LysM-Cre) and control mice (*Mafb^f/f^*) showed that wound healing was significantly delayed in the former. For wounded GFP knock-in mice with GFP inserts in the *Mafb* locus, flow cytometry revealed that their GFP-positive cells expressed macrophage markers. Thus, macrophages express *Mafb* at wound sites. Immunohistochemical (IHC) staining, proteome analysis, and RT-qPCR of the wound tissue showed relative downregulation of *Arg1*, *Ccl12*, and *Ccl2* in *Mafb^f/f^*::LysM-Cre mice. The aforementioned genes were also downregulated in the bone marrow-derived, M2-type macrophages of *Mafb^f/f^*::LysM-Cre mice. Published single-cell RNA-Seq analyses showed that *Arg1*, *Ccl2*, *Ccl12*, and *Il-10* were expressed in distinct populations of MAFB-expressing cells. Hence, the MAFB-expressing macrophage population is heterogeneous. MAFB plays the vital role of regulating multiple genes implicated in wound healing, which suggests that MAFB is a potential therapeutic target in wound healing.

## 1. Introduction

V-maf musculoaponeurotic fibrosarcoma oncogene family, protein B (MAFB) is a member of the Maf transcription factor (TF) family that includes basic-region leucine zipper-type TFs, which bind the Maf recognition element (MARE) by dimerization [1]. MAFB was reported as an oncogene involved in inducing multiple myeloma [2], and Human Protein Atlas (HPA) data indicate that the expression of MAFB is related to the growth of endometrial cancer [3].

MAFB is widely expressed in several tissues and is specifically expressed in the macrophages of the hematopoietic tissue [4]. In particular, HPA data show that MAFB is highly expressed in Kupffer cells in humans [5]. In vitro analyses demonstrated that MAFB is specifically expressed in anti-inflammatory IL-4- or IL-10-induced, bone marrow-derived M2-type macrophages [6]. A previous study showed that MAFB regulates the *C1q* gene, the first protein in the classical pathway, to promote efferocytosis and prevents autoimmune diseases in tissue-resident macrophages [7]. MAFB is also expressed in macrophages associated with various disease conditions. In ischemic stroke, MAFB may regulate the expression of mouse scavenger receptor 1 (*MSR1*) and macrophage receptor with collagenous structure (*MARCO*), thereby reducing the levels of damage-associated molecular patterns (DAMPs) [8]. In atherosclerosis, MAFB prevents the apoptosis of macrophages by regulating the apoptosis inhibitor of macrophage (*AIM*) gene [9]. MAFB also inhibits the self-renewal function of terminally differentiated macrophages [10]. Therefore, MAFB can regulate macrophage function and maintain homeostasis in disease. With these functions, it is possible that MAFB also relates to other diseases and abnormal conditions, such as wound healing.

Repair and homeostasis in response to skin injury are essential. Rapid wound repair is vital in the event of severe trauma. In previous studies, macrophages were removed during wound healing and were empirically demonstrated to be essential for this process [11,12]. Lucas et al. reported that macrophage removal during the inflammatory phase decreased angiogenesis and delayed the formation of the granulation tissue [13]. However, macrophage removal during the proliferative phase resulted in severe bleeding into the wound tissue [13]. These findings suggest that macrophages regulate different cells at each stage of tissue repair and that a lack of cellular response at any stage can delay repair and lead to chronic wounds. Therefore, macrophage function must be tightly regulated. Nevertheless, the mechanisms determining which macrophage subsets are functional at each phase of wound repair remain to be elucidated. Macrophages express cytokines and chemokines that play important roles in wound healing. CCL2 promotes re-epithelialization by recruiting macrophages [14]. CCL12 suppresses inflammation and induces fibrosis [15]. IL-10 suppresses excessive fibrosis during remodeling [16]. Arg1 is a well-known M2 macrophage marker that induces fibrosis during wound healing [17]. However, transcriptional regulation of the aforementioned factors is yet to be clarified. Furthermore, the *TLR2*^−/−^ mice delayed wound healing [18], and HPA indicates that *MAFB* is coexpressed with *TLR2* in humans [5]. However, the expression of these in mice wounds is unknown.

In the present study, mice with specific *Mafb* knockouts in certain macrophage lineages (*Mafb^f/f^*::LysM-Cre) were used to investigate the roles and functions of MAFB in wound healing. We discovered that MAFB was expressed in the granulation tissue and that wound healing was delayed in *Mafb^f/f^*::LysM-Cre mice. Moreover, *Ccl12*, *Arg1*, and *Ccl2* were downregulated in the wound sites by day 3 after skin injury in the *Mafb^f/f^*::LysM-Cre mice. Hence, macrophage MAFB induces genes associated with wound repair and is crucial for proper wound healing.

## 2. Results

### 2.1. MAFB Is Required for Proper Wound Healing

We wounded *Mafb^f/f^* and *Mafb^f/f^*::LysM-Cre mice and measured the wound areas on days 0, 3, 5, 7, and 10 to confirm whether MAFB is related to wound healing (Figure 1A,B). The average wound area in *Mafb^f/f^*::LysM-Cre mice was 1.3, whereas it was 1 in *Mafb^f/f^* mice (*p* = 0.009) on day 3. On day 5, the area in *Mafb^f/f^*::LysM-Cre was 1.4, whereas it was 1 in *Mafb^f/f^* mice (*p* = 0.024). By day 10, several *Mafb^f/f^* mice presented with closed wounds, whereas only two *Mafb^f/f^*::LysM-Cre mice demonstrated wound closure. Hence, wound healing was significantly impaired in the *Mafb^f/f^*::LysM-Cre mice. We used hematoxylin and eosin (H&E) staining to examine the histological differences between *Mafb^f/f^* and *Mafb^f/f^*::LysM-Cre wounds and observed their hallmarks on days 3 and 5 (Figure 1C,D). We found that the areas of the dermis in the H&E sections were significantly larger in the *Mafb^f/f^*::LysM-Cre mice than they were in the *Mafb^f/f^* mice on day 3 (*p* = 0.023; Figure 1D). We used Masson’s trichrome staining to measure the collagen area in the granulation tissue and found that it was significantly decreased in the *Mafb^f/f^*::LysM-Cre mice relative to that in the *Mafb^f/f^* mice on day 5 (Figure 1E). Therefore, MAFB was required for prompt and timely wound healing and was implicated in collagen accumulation.

### 2.2. MAFB Expressed on Ly6C^+^ Macrophages in Granulation Tissue

Immunostaining was performed on the day 3 wounds of *Mafb^+/GFP^* mice to confirm MAFB expression. GFP expression was higher in the granulation tissue (Figure 2A,B) than in the dermis (Figure 2C). To confirm whether the GFP-expressing cells were macrophages, we performed FACS analysis on the day 3 wounds on *Mafb^+/GFP^* mice. To this end, we used the myeloid marker antibody CD11b and the macrophage marker antibodies CD204, Mac2, and F4/80 (Figure 2D) and observed the cell populations coexpressing GFP with all the macrophage markers. We performed FACS analysis on the day 3 wounds of *Mafb^+/GFP^* mice to identify the GFP-expressing cells. For this purpose, we used antibodies against CD11b, Gr-1, and Ly6C (Figure 2E). The green dots indicate GFP^+^ cells. Whereas the Gr-1^−^, CD11b^+^, Ly6C^+^ monocyte-derived macrophages expressed GFP, the CD11b^+^, neutrophil marker Gr-1^+^ cells did not. Hence, MAFB was expressed in the granulation tissue and, more precisely, in the macrophages around the wounds.

### 2.3. MAFB Expression in Macrophages Is Related to Their Recruitment in Wounds

FACS analysis was performed on day 0, 3, 5, and 7 wounds to compare *Mafb^f/f^* and *Mafb^f/f^*::LysM-Cre mice in terms of the numbers and dynamics of the cells in their wounds during healing. The neutrophils (CD11b^+^, Gr-1^+^ cells) and monocytes/macrophages (CD11b^+^, Gr-1^−^ cells) were enumerated and are displayed in Appendix A. Gating strategies are shown in Appendix A. The numbers of whole monocytes/macrophages were low under normal conditions (day 0) but gradually increased until day 7 after injury (Appendix A). The monocyte/macrophage population was divided into the CD11b^high^ and CD11b^low^ subpopulations (Appendix A). The latter were further subdivided into the CD11b^high^, Ly6C^−^, CD11b^high^, Ly6C^+^, and CD11b^low^, Ly6C^+^ cell subpopulations (Figure 3A). Relative to day 0, the cell numbers in all populations increased by day 3. However, only the CD11b high, Ly6C^+^ cells peaked at day 3, and they decreased thereafter. In contrast, the numbers of the CD11b^high^, Ly6C^−^ cells and CD11b^low^, Ly6C^+^ cells continued to increase after day 3 and reached their peak by day 7. The foregoing experiments demonstrated cell dynamics and numbers during wound healing. By day 5, the CD11b^high^, Ly6C^+^ cells were significantly lower in the *Mafb^f/f^*::LysM-Cre mice than in the *Mafb^f/f^* mice. Thus, MAFB might recruit monocytes/macrophages.

### 2.4. Mafb Deficiency Is Associated with CCL12, CCL2, and Arg1 Downergulation in Wounds

The effects of *Mafb* deletion on the number of cells were evident by day 5. Therefore, we used kits to measure the proinflammatory and anti-inflammatory cytokines and angiogenesis-related proteins in the wound tissues on day 3 (Appendix A). Figure 4A is an enlarged photograph of the membrane. CCL12 and CXCL10 were slightly expressed in the *Mafb^f/f^* mice but were not expressed at all in the *Mafb^f/f^*::LysM-Cre mice. The IL-10 and CCL2 expression levels were lower in the *Mafb^f/f^*::LysM-Cre mice than in the *Mafb^f/f^* mice. Several differentially expressed genes detected between the *Mafb^f/f^* mice and the *Mafb^f/f^*::LysM-Cre mice in the proteome analysis were selected and their mRNA levels in day 3 wound tissue were quantitated using RT-qPCR (Figure 4B). *Ccl12* was significantly decreased and *Ccl2* tended to decrease in the *Mafb^f/f^*::LysM-Cre mice. However, the expression levels of *Cxcl10*, *Il-10*, and the other genes did not differ between the *Mafb^f/f^* and *Mafb^f/f^*::LysM-Cre mice. Moreover, the expression levels of the proinflammatory genes, *Il-12b* and *Nos2*, did not differ between the *Mafb^f/f^* and *Mafb^f/f^*::LysM-Cre mice. Here, wound tissue samples also contain various cells other than macrophages. Therefore, gene expression in other cells may mask differences in the gene expression of macrophages. For this reason, we conducted RT-qPCR on bone marrow-derived M2 macrophages induced with M-CSF, IL-4, and IL-13 in *Mafb^f/f^* and *Mafb^f/f^*::LysM-Cre mice. *Ccl12* and *Ccl2* were downregulated both in the *Mafb^f/f^*::LysM-Cre mice and the wound tissue. In contrast, the *I**l-10, Arg1*, and *c-Maf* expression levels did not differ among wound tissues but significantly differed in M2 macrophages in vitro (Figure 4C). It has been reported that *c-Maf* and *Mafb* are mutually compensatory. However, this compensation was not apparent under the M2 condition. *Il-10* is the direct target of *c-Maf* and was consistently downregulated in *Mafb^f/f^*::LysM-Cre mice. Therefore, MAFB induces *Ccl12*, *Arg1*, *c-Maf*, and *Il-10*. In *Mafb^f/f^*::LysM-Cre mice, *Ccl12* expression was 2.5- and 100-fold lower in wound tissue and bone marrow-derived M2 macrophages, respectively (Figure 4B,C). Hence, *Mafb* induces *Ccl12* expression in both wound tissue and macrophages in vitro.

In a previous study, *Arg1^f/f^*::Tie2-Cre mice presented with reduction in collagen accumulation and delayed wound healing [17]. Here, Masson’s trichrome staining showed that collagen accumulation decreased on day 5 in *Mafb^f/f^*::LysM-Cre mice (Figure 1E). We confirmed *Arg1* expression in day 3 and 5 granulation tissue using immunostaining (Figure 4D). The ratios of Arg1-positive area per granulation tissue area were significantly reduced in the *Mafb^f/f^*::LysM-Cre mice on days 3 and 5. The expression levels of the macrophage marker galectin-3 (Mac2) did not differ between *Mafb^f/f^* and *Mafb^f/f^*::LysM-Cre mice (Appendix A). Therefore, Arg1 expression may have specifically decreased in the macrophages that infiltrated the granulation tissue through *Mafb* deletion. Thus, MAFB may induce *Ccl12*, *Ccl2*, and *Arg1* during wound healing.

### 2.5. Mafb-Positive Macrophages Show Heterogeneous Gene Expression

Both tissue-resident and infiltrated macrophages are intricately involved in wound healing [19]. We aimed to identify the cell population expressing genes that MAFB could regulate. To this end, we analyzed the data for the day 5 whole wound tissue, as well as for the macrophages infiltrating the wounds [20,21]. In the latter case, we traced the RFP^+^ monocytes injected before harvesting the wound tissue (Figure 4E,F).

*Cd68*^+^ cells were detected in the whole wound tissue (GEO accession no. GSE140512) (Appendix A) [20]. It was assumed that cells with comparatively higher *Cd68* expression levels were macrophages (Appendix A). We also measured *Mafb*, *c-Maf*, *Arg1*, *Il-10*, *Ccl2*, *Ccl12*, and *Cxcl10* expression (Figure 4E). *Mafb* was expressed mainly in clusters 0–3. Expression of *c-Maf* was slightly lower than that of *Mafb*. Nevertheless, the *c-Maf*^+^ and *Mafb^+^* clusters were similar (Figure 4E). *Arg1* and *Ccl12* were expressed exclusively in the *Mafb*^+^, no. 0 and no. 2 clusters, respectively. The expression levels of *Il-10* and *Cxcl10* were low. Whereas most *Il-10^+^* cells expressed *Mafb*, some *Cxcl10^+^* cells did not. Overall, *Ccl2* expression was weak and the main population resembled that of *Ccl12*. Hence, the *Mafb*^+^ cells also expressed *c-Maf*, *Arg1*, *Ccl12*, *Ccl2*, *Il-10*, and *Cxcl10*. However, the gene populations were heterogeneous.

Data for the day 4 and 14 wound macrophages were then analyzed (GEO accession no. GSE183489) (Figure 4F) [21]. *Mafb* was not expressed in cluster 9 and the *Ccl2* population resembled the *Mafb* population. The latter had a slightly higher expression level than that of *c-Maf*. However, most *c-Maf*^+^ cells expressed *Mafb* mainly in clusters 2, 4, and 6 (Figure 4F). *Ccl12*, *Arg1*, *Il-10*, and *Cxcl10* exhibited different expression patterns. *Ccl12* was slightly expressed in all clusters, whereas *Arg1* was expressed mainly in clusters 1, 3, and 4 and *Il-10* was expressed mainly in clusters 0, 2, and 6. Therefore, the gene populations were not identical and most of them expressed *Mafb*. These findings were similar to those for the whole wound tissue analysis (Figure 4E). *Ccl2* expression differed between the whole tissue and infiltrated wound macrophages. *Ccl2* expression was relatively low in the *Cd68*^+^ macrophages of whole wound tissue (Figure 4E). In the infiltrated wound macrophages, however, *Ccl2* was comparatively upregulated (Figure 4F). Thus, *Ccl2* was expressed mainly by infiltrated macrophages in wounds. Additionally, because TLR2 has been reported to be important for the wound healing process [18] and its gene is coexpressed with *MAFB* in humans [5], we confirmed the expression of *Tlr2* in mice wound scRNA-seq data. The expression pattern of *Tlr2* was more similar to that of *Mafb* than that of *Tlr4*. This may suggest the possibility of a relationship between the TLR2 pathway and MAFB (Appendix A). The foregoing findings indicate that not all genes regulated by MAFB were expressed within the same population and that MAFB-expressing cells constituted a heterogeneous population. Thus, MAFB may affect wound healing via several genes.

## 3. Discussion

To the best of our knowledge, the present study is the first to investigate the role of MAFB in skin wound healing in vivo. At 3–5 days after the lesions were induced, the wounds were relatively larger and the collagen areas had decreased in the *Mafb^f/f^*::LysM-Cre mice compared with that in the *Mafb^f/f^* mice. At this time point, the *Ccl12*, *Ccl2*, and *Arg1* expression levels were lower in the wound tissue- and bone marrow-derived M2-type macrophages (Figure 4B–D). Therefore, MAFB induces *CCL12*, *CCL2*, and *Arg1* in the macrophages mediating tissue repair during wound healing. It was reported that CCL12 is a ligand of CCR2. While CCL12 occurs only in mice, it has high homology to human CCL2 [15,22,23]. Moore et al. showed that CCR2-mediated signaling recruited fibroblasts to the pulmonary alveoli in a lung fibrosis model. Moreover, pulmonary fibrosis was suppressed by neutralizing CCL12 but not CCL2 [15,24]. Here, we observed that collagen decreased in the *Mafb^f/f^*::LysM-Cre mice on day 5 after wounding (Figure 1E). Hence, CCL12 and CCL2 may promote fibrosis during wound healing. *Mafb* deletion in macrophages decreased the numbers of Gr-1^−^, CD11b^+^, Ly6C^+^ cells on day 5 (Figure 3). Thus, MAFB may participate in monocyte/macrophage recruitment. CCR2 signaling induces Ly6C^hi^ monocyte/macrophage recruitment to wound sites and promotes wound repair [25]. In a pulmonary fibrosis model, CCL12 was important for the recruitment of exudate macrophages and Ly6C^hi^ monocytes to the lungs [23]. It is, therefore, possible that CCL12 and CCL2 were induced by MAFB on day 5 and promoted Ly6C^hi^ monocyte/macrophage recruitment to the wound sites. Nevertheless, the initial influx of Ly6C^hi^ monocytes/macrophages into the wound occurred on day 3 and there were no significant differences between the *Mafb^f/f^* and *Mafb^f/f^*::LysM-Cre mice at that time. Thus, it is improbable that the observed decreases in the number of cells on day 5 had any significant effect on delayed wound healing.

Arg1 also contributed to the observed decrease in collagen in the *Mafb^f/f^*::LysM-Cre mice. This hydrolytic enzyme degrades arginine into urea and ornithine. The latter may be converted to proline required for collagen synthesis and polyamines, which are essential for cell growth and differentiation. All three processes are vital to wound healing [17,26]. There were no significant differences between *Mafb^f/f^* and *Mafb^f/f^*::LysM-Cre mice in terms of *Arg1* mRNA expression in their wound tissues (Figure 4B). However, an in vitro analysis of BM-derived M2 macrophages showed that *Arg1* mRNA expression was significantly reduced in *Mafb^f/f^*::LysM-Cre mice (Figure 4C). Further, histological analysis showed that Arg1 expression was significantly decreased in the granulation tissues of *Mafb^f/f^*::LysM-Cre mice on days 3 and 5 (Figure 4D). Arg1 is also expressed at high levels in keratinocytes and especially in those at wound edges [27]. The edges of the wound tissues used in RT-qPCR (Figure 4B) probably contained keratinocytes. For this reason, *Arg1* mRNA expression might not have been rigorously evaluated in the granulation tissue. The foregoing results suggest that MAFB may induce Arg1 and, by extension, collagen production in granulation tissue.

The mRNA expression levels of *c-Maf* and *Il-10* did not differ in the wound tissue. Nevertheless, *Mafb* deletion downregulated *c-Maf* and *Il-10* in in vitro BM-derived M2 macrophages. Keratinocytes, T cells, and monocytes/macrophages express c-Maf [28,29]. T cells and monocytes/macrophages produce IL-10 [30]. Keratinocytes produce IL-10 after wounding [30]. RT-qPCR analysis of wound tissue (Figure 4B) may therefore detect *c-Maf* and *Il-10* induced by cells other than macrophages in the granulation tissue. In the present study, we found that MAFB induces *c-Maf* whereas M-CSF, IL-4, and IL-13 induce *Il-10* in M2 macrophages. As c-Maf might promote IL-10 expression in macrophages [31], MAFB could regulate *Il-10* expression via *c-Maf* regulation.

An evaluation of previously reported single-cell RNA sequencing (scRNA-Seq) data on whole wound tissue and infiltrated wound macrophages revealed the relationship between *Mafb* and its downstream genes. The scRNA-Seq data disclosed that *Mafb* is expressed in most clusters whereas the other cytokines are only expressed in a few of them. The preceding data indicate that *Mafb*^+^ macrophages are highly heterogeneous. The roles of *Mafb*^+^ macrophages may vary depending on their relative gene expression levels. We found that macrophage MAFB induces *C1q* [7]. C1q is a recognition signal that triggers the classical complement pathway, promotes the clearance of apoptotic cells, and contributes to cytokine/chemokine induction, depending on the receptors involved [32]. *C1qa*^−/−^ mice presented with defective vascularization. However, local C1q application rescued this process [33]. These discoveries indicate that MAFB may contribute to angiogenesis and cytokine/chemokine induction and that it has functional diversity. The findings of the present work suggest that MAFB participates in wound closure, collagen production, and monocyte/macrophage recruitment by regulating the expression of *Ccl12*, *Ccl2*, *Arg1*, and other genes in various macrophage populations. In this manner, MAFB promotes integrated wound healing. Future studies on MAFB functions in each cluster will help elucidate the roles of macrophages in wound healing through scRNA-seq using *Mafb*-deficient wound tissue.

Previously, we showed that MAFB is regulated by nuclear receptor transcription factors. In vitro results show that agonists of PPARs, LXRs, RARs, and glucocorticoid receptors (GRs) increase MAFB expression in macrophages [7,9]. These agonists have been shown to induce immunosuppression and fibrosis during focal segmental glomerulosclerosis (FSGS), and MAFB may be an important hub in these signals [34]. However, we could not confirm whether these agonists are also functional during wound healing in this study. Further, *Mafb*^−/−^ mice embryos showed impaired epidermal differentiation, indicating the importance of MAFB in keratinocytes for differentiation of keratinocytes [35]. In this study, we found that macrophage-specific, *Mafb*-deficient mice also showed impaired epithelialization, but the mechanism remains unclear. Furthermore, the roles of MAFB in other kinds of injuries, such as burns or wounds in other organs, are unknown. This study is the first step toward showing the potential of MAFB as a new therapeutic target. Further analysis of the transcriptional regulation of the MAFB gene may provide support for the applicability of MAFB for tissue repair under several conditions and in the field of plastic surgery.

## 4. Materials and Methods

### 4.1. Mice

*Mafb* GFP knock-in mice (*Mafb* heterozygous (*Mafb^gfp/+^*)) with a C57BL/6J background were used in the present study. Mouse genomic DNA was extracted from the tail tips and the genotypes were characterized using PCR. The sequences of the genotyping primers were previously reported [9]. For the *Mafb* conditional knockout mice, *Mafb* was flanked by a loxP element with a neomycin-resistant gene via homologous recombination in C57BL/6 background ES cells [36]. *Mafb^f/f^* mice were mated with LysM-Cre mice (Jackson Laboratory, Hancock, ME, USA) to delete *Mafb* in the macrophage lineage. *Mafb* expression was under the control of the endogenous Lyz2 promoter [37]. The mice were maintained under specific pathogen-free conditions in a laboratory animal resource center at the University of Tsukuba, Japan.

### 4.2. Wounding and Wound Collection

The back hair of male mice aged 8–10 weeks was shaved and the mice were anesthetized with isoflurane. Two full-thickness wounds were made ≥5 mm apart on the denuded back skin with a 5 mm diameter biopsy punch (Kai Industries Co., Ltd., Gifu, Japan, Cat. No. BP-50F). Analyses were performed on 3, 5, 7, and 10 d after wounding. The mice were sacrificed by CO_2_ inhalation and the wounds were collected with an 8 mm diameter biopsy punch (Kai Industries Co., Ltd., Gifu, Japan, Cat. No. BP-80F).

### 4.3. Wound Area Quantification

The wound areas were evaluated on days 0, 3, 5, 7, and 10 after the injuries were inflicted to determine the wound closure rates. Rulers were positioned adjacent to the wounds and photographs were taken and analyzed using Photoshop software (https://www.adobe.com/products/photoshop.html, accessed on 17 August 2022). Data were expressed as percentages of the initial wound area.

### 4.4. Histological Analysis

Wound tissues were fixed in Mildform 10N (Wako, Osaka, Japan, Cat. No.133-10311) overnight at 4 °C and paraffinized. A paraffinized tissue block was sliced with 3 μm thickness using a Sliding Microtome SM2000R (Leica Biosystems, Wetzlar, Germany) and processed for H&E, Masson’s trichrome, and immunohistochemical (IHC) staining with anti-GFP (Aves Labs, Davis, CA, USA, Cat. No. GFP-1020), anti-Arg1 (Santa Cruz Biotechnology, Dallas, TX, USA, Cat. No. sc-271430), and anti-Mac2 (CEDARLANE, Burlington, ON, Canada, Cat. No. CL8942AP) antibodies. Antigen retrieval (Dako, Santa Clara, CA, USA, Cat. No. S1699) was used exclusively for GFP staining. For GFP IHC staining, the sections were blocked with a mixture of 5% (*v*/*v*) goat serum (Gibco, Carlsbad, CA, USA, Cat. No. 16210064), 1% (*v*/*v*) bovine serum albumin (BSA) (Sigma-Aldrich, St. Louis, MO, USA, Cat. No. A7906-100G), and 0.1% (*v*/*v*) Triton (Wako, Osaka, Japan, Cat. No. 162-24755) in phosphate-buffered saline (PBS). For Arg1 IHC staining, the sections were blocked with 10% (*v*/*v*) donkey serum in PBS. For Mac2 IHC staining, the sections were blocked with 5% (*v*/*v*) BSA in PBS. Blocking was performed at 4 °C for 1 h. The sections were then incubated with primary anti-GFP (1:500 dilution), anti-Arg1 (1:200 dilution), and anti-Mac2 (1:200 dilution) antibodies at 4 °C overnight. The sections were subsequently washed and incubated with Goat anti-chicken Alexa Fluor 488 (Life-technologies, Carlsbad, CA, USA, Cat. No. A11039), Donkey anti-mouse Alexa Fluor 546 (Invitrogen, Carlsbad, CA, USA, Cat. No. A10036), and Chicken anti-rat Alexa Fluor 488 (Invitrogen, Carlsbad, CA, USA, Cat. No. A21470) secondary antibodies at 20–25 °C for 1 h.

### 4.5. Flow Cytometry

Round skin wounds 8 mm in diameter were collected and cut with scissors into ~2 mm squares. To isolate the cells, the wound sections were placed in RPMI buffer (0.05% (*w*/*v*) DNase I (Roche, Basel, Switzerland, Cat. No. 11284932001) plus 10% (*v*/*v*) fetal bovine serum (FBS) (Gibco, Carlsbad, CA, USA, Cat. No. 10270-106)) containing Liberase^TM^ TL Research Grade (Roche, Basel, Switzerland, Cat. No. 5401020001) (0.5 mg/mL) and shaken at 180 r/min at 37 °C for 20 min. The cells were resuspended in FACS buffer (5% (*v*/*v*) FBS plus 0.5 mM EDTA) and stained with the aforementioned antibodies. DAPI (4′,6-diamidino-2-phenylindole; Dojindo Laboratories, Kumamoto, Japan) was used to stain the dead cells. CytExpert software Version 2.4 (https://www.beckman.com, accessed on 17 August 2022) (Beckman Coulter, Brea, CA, USA) was used for analysis.

### 4.6. Proteome Analysis

Proteome Profiler Mouse Cytokine Array Kit, Panel A (R&D Systems, Minneapolis, MN, USA, Cat. No. ARY006) and Proteome Profiler Mouse Angiogenesis Array Kit (R&D Systems, Minneapolis, MN, USA, Cat. No. ARY015) were used to analyze the cytokines and proteins in the wounds. Briefly, granulation tissues were collected under a stereomicroscope S6E (Leica Microsystems, Wetzlar, Germany) from the day 3 wounds. The tissues were homogenized with a Micro Smash MS-100R Cell Disruptor (Tomy Seiko Co., Ltd., Tokyo, Japan) in PBS with protease inhibitors. Triton X-100 (Wako, Osaka, Japan, Cat. No. 162-24755) was added to get a final concentration of 1%. Protein lysates (300 mg) were quantitated using the Bicinchoninic Acid Assay (Bio Rad Laboratories, Hercules, CA, USA) according to the manufacturer’s instructions.

### 4.7. qRT-PCR

Total RNA was collected with an Isogen Kit (Nippon Gene, Tokyo, Japan, Cat. No. 311-02501). The cDNA was synthesized with a QuantiTect Reverse Transcription Kit (Qiagen, Hilden, Germany, Cat. No. 205313). Gene expression levels were determined using RT-PCR performed on a Thermal Cycler Dice Real Time System Single TP850 (Takara Bio Inc., Shiga, Japan) and a THUNDERBIRD^®^ SYBR^®^ qPCR Mix (TOYOBO Co., Ltd., Osaka, Japan, Cat. No. QPS-201). The mRNA levels were normalized to that of Hprt. The primer nucleotide sequences are shown in Appendix A.

### 4.8. Bone Marrow Macrophage Culture

Bone marrow cells (1 × 10^6^) were suspended in 3 mL medium (10% (*v*/*v*) FBS plus 1% (*w*/*v*) Pen-Strep (Sigma-Aldrich, St. Louis, MO, USA)), seeded in a six-well dish, and cultured with 30 ng/mL M-CSF (R&D Systems, Minneapolis, MN, USA, Cat. No. 416-ML-500) at 37 °C for 3 d. The existing medium was then replaced with 3 mL fresh medium containing 30 ng/mL M-CSF. Thereafter, 20 ng IL-4 (R&D Systems, Minneapolis, MN, USA, Cat. No. 404-ML-010) and 20 ng IL-13 (R&D Systems, Minneapolis, MN, USA, Cat. No. 413-ML-005) were added to the medium 5 d after seeding. The cells were then collected the following day with an Isogen Kit.

### 4.9. Analysis of scRNA-Seq Data

Single-nuclei RNA-Seq data for wound and non-wound macrophages were analyzed. CD45^−^ and CD31^−^ cells were isolated using flow cytometry from dorsal skin wound beds on day 5. The characteristics of cells with CD68 expression > 0 were then determined. Data processing and analysis were performed with Scanpy (https://github.com/scverse/scanpy, accessed on 17 August 2022). UMAP (https://github.com/lmcinnes/umap accessed on 17 August 2022) and gene expression analyses were subsequently performed. The UMI count was the expression matrix unit (Shook et al., GSE140512) [20]. The raw single-nuclei RNA-Seq data were downloaded from Gene Expression Omnibus (GEO) and then processed with Scanpy. Scanpy performs normalizations and analyzes and generates figures. Cells with <200 or >2500 unique feature counts and >5% mitochondrial counts were filtered out. After the quality control, the cells were subjected to downstream analysis. They were log-normalized and their highly variable genes were identified and subjected to a principal component analysis (PCA), which reduces dimensionality and reveals the main variations in data. A neighborhood graph was then computed, plotted with UMAP, and colored with Leiden. Genes associated with wound healing were selected and their expression levels and distribution were displayed in the macrophage clusters.

Single-nuclei RNA-Seq of mouse wound macrophages was performed. Wounds collected 4 or 14 d after the injury and monocyte transfusion on days 2 or 12 were pooled and analyzed using single-nuclei RNA-Seq analysis. The raw single-nuclei RNA-Seq data were obtained from GEO and processed with Scanpy. Scanpy performs log-normalization and highly variable genes were identified and subjected to PCA. A neighborhood graph was then computed, plotted with UMAP, and colored with Leiden. Genes associated with wound healing were selected and their expression levels and distribution were displayed in the macrophage clusters (Willenborg et al., GSE183489) [21].

### 4.10. Statistical Analyses

Data are means ± S.E.M. Significant differences between group pairs were analyzed by Student’s *t*-test using Microsoft Excel. Differences were considered statistically significant at *p* < 0.05.

## Figures and Tables

**Figure 1 ijms-23-09346-f001:**
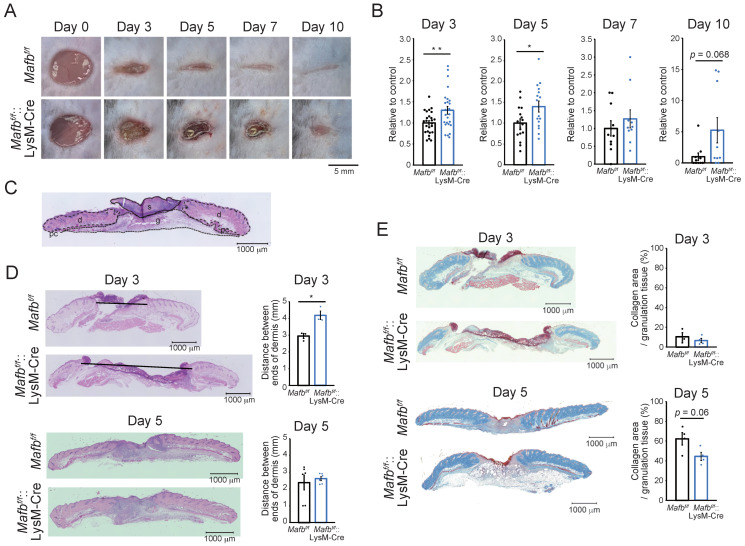
Wound healing impairment in *Mafb^f/f^*::LysM-Cre mice. (**A**) Representative photographs of the wounds in *Mafb^f/f^* and *Mafb^f/f^*::LysM-Cre mice on days 0, 3, 5, 7, and 10 post-injury. (**B**) The wound areas were analyzed using Photoshop on days 0, 3, 5, 7, and 10. The wound area at day 0 was set as 100%. The wound areas in the *Mafb^f/f^* mice were normalized to the mean % of wounds in *Mafb^f/f^* mice (control). The wound areas in the *Mafb^f/f^*::LysM-Cre mice were also normalized to the mean % of their wound areas and analyzed. (**C**) Hematoxylin and eosin (H&E) staining of various wound tissues (s: scab, d: dermis, g: granulation tissue, pc: panniculus carnosus). (**D**) Tissues at the wound sites were harvested on days 3 and 5 and stained with H&E. Graph showing the area of the dermis. (**E**) Masson’s trichrome staining of the wound sections was performed on days 3 and 5. Graph showing the % collagen area per granulation tissue area. Data are means ± S.E.M; * *p* < 0.05, ** *p* < 0.01, Student’s *t*-test. Each dot represents data for a single mouse.

**Figure 2 ijms-23-09346-f002:**
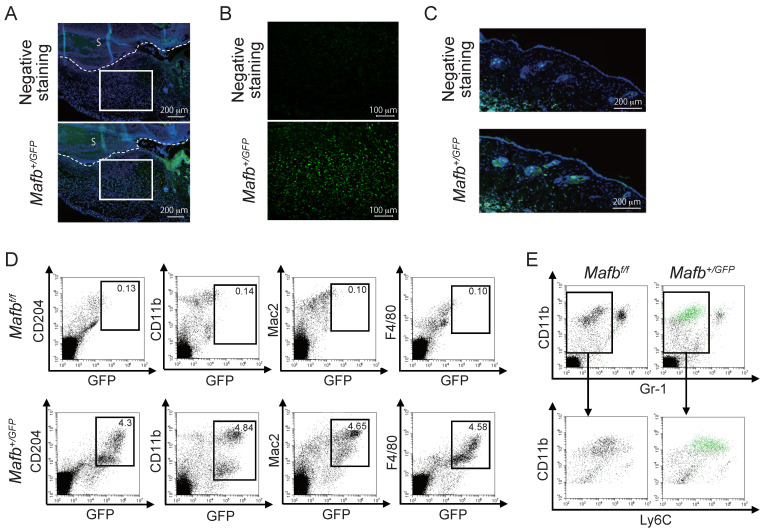
MAFB is expressed in granulation tissue. (**A**) GFP in the day 3 wounds of *Mafb^+/GFP^* mice was immunostained and observed under a microscope at ×100 magnification. (**B**) Granulation tissue enclosed in the square in (**A**) was observed at ×200 magnification. (**C**) GFP expression in the dermis was observed in same tissue as in A. (**D**) Macrophages in the day 3 wounds of *Mafb^+/GFP^* mice were stained with CD204, CD11b, Mac2, and F4/80 and analyzed using flow cytometry. (**E**) GFP-expressing cells (green dots) in the day 3 wounds are shown. Each experiment used *n* = 2–4 *Mafb^+/GFP^* mice. Representative data are shown.

**Figure 3 ijms-23-09346-f003:**
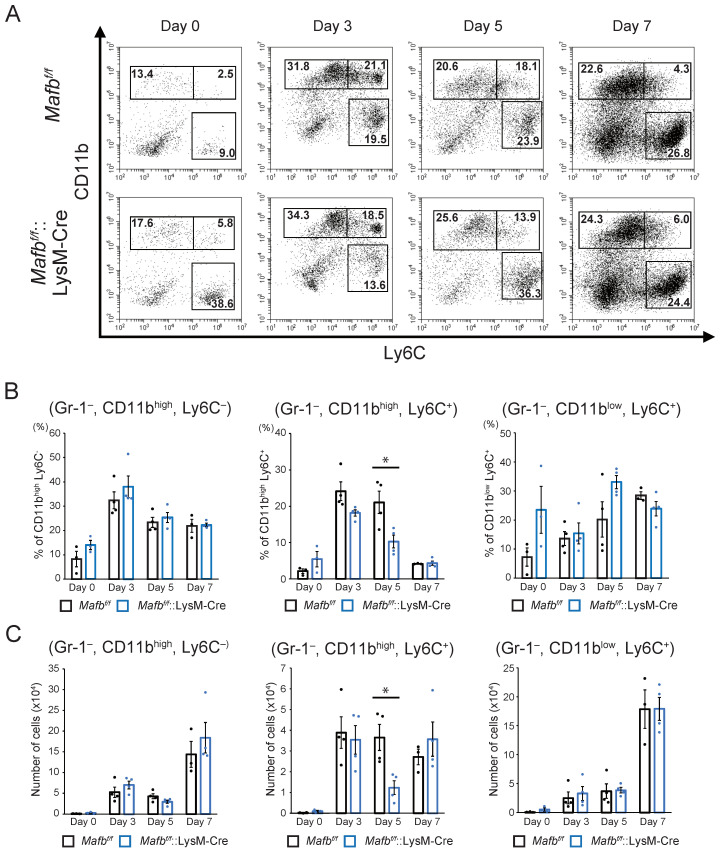
Numbers of monocyte-derived cells decreased in the *Mafb^f/f^*::LysM-Cre mouse wounds. (**A**) Cells in the day 0, 3, 5, and 7 wounds were stained with anti-Gr-1, anti-CD11b, and anti-Ly6C antibodies and analyzed using flow cytometry. Dead cells and neutrophils were identified using DAPI, and CD11b and Gr-1 staining, respectively. Representative dot plots are shown for each day. (**B**) The % of each cell population in A per live Gr-1^−^ cells is shown. (**C**) The absolute numbers of cells in each population in A were calculated from the overall number of cells taken from the tissues and the % of each cell population. Data are means ± S.E.M; * *p* < 0.05, Student’s *t*-test. For each day, *n* = 3–4 for *Mafb^f/f^* and *Mafb^f/f^*::LysM-Cre mice. Each dot represents data for a single mouse.

**Figure 4 ijms-23-09346-f004:**
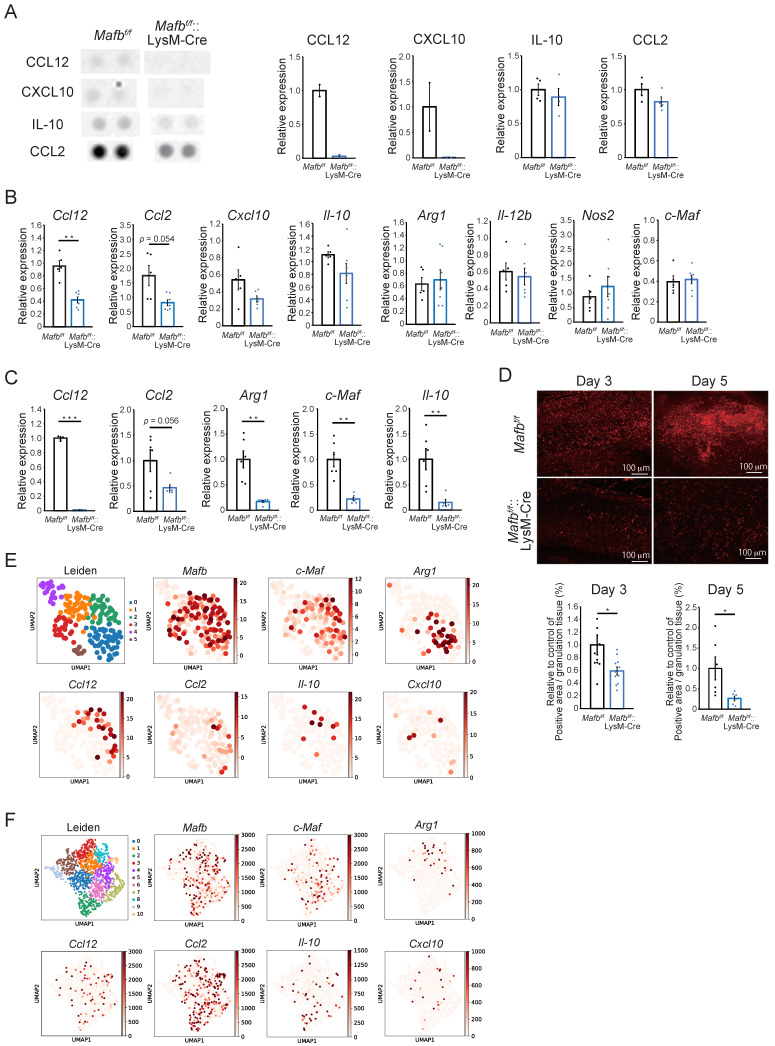
Expression levels of genes related to wound healing were decreased by *Mafb* deletion in macrophages. (**A**) Cytokine arrays were performed on the day 3 wound tissue. CCL12 and CXCL10 expression in *Mafb^f/f^* and *Mafb^f/f^*::LysM-Cre mice was analyzed with a Proteome Profiler Mouse Cytokine Array Kit, Panel A (R&D Systems, Inc., Minneapolis, MN, USA) (*n* = 2 *Mafb^f/f^* and *Mafb^f/f^*::LysM-Cre mice). IL-10 and CCL2 were detected with a Prolactin by Proteome Profiler Mouse Angiogenesis Array Kit (R&D Systems, Minneapolis, MN, USA) (*n* = 4 *Mafb^f/f^* and *Mafb^f/f^*::LysM-Cre mice). (**B**) The mRNA expression levels in the day 3 wound tissues of *Mafb^f/f^* (*n* = 5) and *Mafb^f/f^*::LysM-Cre mice (*n* = 7) were analyzed using RT-qPCR. (**C**) Relative mRNA expression levels in macrophages derived from the bone marrow of *Mafb^f/f^* (*n* = 6) and *Mafb^f/f^*::LysM-Cre mice (*n* = 6) and induced with M-CSF, IL-4, and IL-13 were analyzed using RT-qPCR. (**D**) Arg1 immunostaining was performed on the day 3 and 5 wound tissues. The % of Arg1-positive pre-granulation tissue area was analyzed. Data for two independent experiments relative to the control are shown in the graph. (**E**) Published scRNA-seq data for the macrophage lineages sampled from the day 4 and 14 wounds were analyzed with UMAP (https://github.com/lmcinnes/umap, accessed on 17 August 2022). (**F**) Published scRNA-seq data for day 5 wounds were also analyzed with UMAP. CD68^+^ cells were assumed to be macrophages and their gene expression levels were analyzed. Data are means ± S.E.M; * *p* < 0.05, ** *p* < 0.01, *** *p* < 0.001 according to Student’s *t*-test. Each dot represents the data for a single mouse.

## Data Availability

The data presented in this article or in Appendix A are available from the corresponding author upon reasonable request.

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
