# Peer review of "Macrophage-Specific, Mafb-Deficient Mice Showed Delayed Skin Wound Healing"

_ijms, 2022, doi:10.3390/ijms23169346_

Round 1
Reviewer 1 Report
Inoue et al. showed that wound healing was significantly delayed in the MAFB-deficient mice (Mafbf/f::LysM-Cre) compared to control mice (Mafbf/f). This is an interesting finding from the researchers explicitly investigating the role of Mafb in macrophages. The manuscript is suitable for publication after minor revisions.
Title: The abnormal skin wound healing does not precisely demonstrate the findings of your research, and I would like to suggest mentioning that the wound healing is delayed in Mafb-deficient mice.
Abstract: The keywords are not sufficient. You may add other keywords such as the significantly altered genes and proteins related to the wound-healing process.
Introduction:
1- The complete form of a gene or protein should be mentioned in the text while introduced, such as MAF bZIP transcription factor B for MAFB.
2- MAFB has the highest mRNA expression in Kupffer cells of the liver, which is higher than other macrophages. Please mention in the introduction and cite. Moreover, the authors said that "MAFB is also expressed in macrophages associated with various disease conditions." I want to suggest some of these diseases and the diseases that are not related to macrophages, but MAFB is overexpressed. Moreover, the expression of MAFB has been linked to TLR2 expression. I would like to see this in your introduction, as well.
3- MAFB has been introduced as a prognostic marker in endometrial cancer. Could you please give some information about the role of MAFB in malignancies, including endometrial cancer?
Results:
1- "It was reported that MAFB may be highly expressed in anti-inflammatory macro-phages that suppress excessive inflammation and promote angiogenesis and cell proliferation [3,15]. Therefore, we hypothesized that MAFB contributes to anti-inflammatory macrophage function during wound healing." These phrases should be relocated to the discussion and not related to results. Moreover, I would like to suggest presenting the quantitative results in the 2.1 section with numbers and p-values.
2- In figure 1A, please show the diameters with a scale to better correlate with Figure 1B. Figure 1D top does not have a scale bar and the unit of distances is not mentioned.
Discussion:
Limitations of the study should be mentioned.
Methods:
How did you make sure that after cutting mice and making wounds, the other mice would not harm them?
Author Response
Reviewer 1
Inoue et al. showed that wound healing was significantly delayed in the MAFB-deficient mice (Mafbf/f::LysM-Cre) compared to control mice (Mafbf/f). This is an interesting finding from the researchers explicitly investigating the role of Mafb in macrophages. The manuscript is suitable for publication after minor revisions.
Title: The abnormal skin wound healing does not precisely demonstrate the findings of your research, and I would like to suggest mentioning that the wound healing is delayed in Mafb-deficient mice.
Response: Thank you for your valuable suggestion. We have revised the title accordingly.
Abstract: The keywords are not sufficient. You may add other keywords such as the significantly altered genes and proteins related to the wound-healing process.
Response: Thank you for your suggestion. We have added new keywords.
Introduction:
1- The complete form of a gene or protein should be mentioned in the text while introduced, such as MAF bZIP transcription factor B for MAFB.
Response: Thank you for important comment. Because MAFB is identified as a homolog of v-maf oncogene cloned from the avian musculoaponeurotic fibrosarcoma virus AS42, we have mentioned “V-maf musculoaponeurotic fibrosarcoma oncogene family, protein B (MAFB)” in the Introduction section.
2- MAFB has the highest mRNA expression in Kupffer cells of the liver, which is higher than other macrophages. Please mention in the introduction and cite. Moreover, the authors said that "MAFB is also expressed in macrophages associated with various disease conditions." I want to suggest some of these diseases and the diseases that are not related to macrophages, but MAFB is overexpressed. Moreover, the expression of MAFB has been linked to TLR2 expression. I would like to see this in your introduction, as well.
Response: Thank you for your pertinent suggestions. We checked the human protein atlas (HPA, https://www.proteinatlas.org) and confirmed that the expression of human MAFB is the highest in Kupffer cells, which suggests the role of MAFB in maintaining homeostasis by clearing harmful factors, such as apoptotic cells or damage associated molecular processes (DAMPs), in the liver. We have included this information in the Introduction section.
With regard to diseases that are not related to macrophages but are associated with the overexpression of MAFB, multiple myeloma (MM) is one of the famous diseases in which there is an overexpression of the large MAF family. MAFB has been detected in samples from MM patients. As a result of the t(14;20) translocation, Mafb gene is translocated to the immunoglobulin heavy chain locus; consequently, MAFB is strongly expressed and induces malignancy by regulating the genes involved in proliferation, such as CYCLIN D2. Because all this information is not directly related to our study, we have summarized it in a sentence in the Introduction section.
Regarding the relationship between TLR2 and MAFB, surprisingly, the HPA data show that the expression pattern of TLR2 in macrophage clusters is most related to that of MAFB. We were unaware of this fact, and probably it has not been reported earlier. We have added this information in the Introduction section. To further investigate whether the same is true in the mouse model, we compared the expression of TLR2 and MAFB using single cell RNA data from wound healing samples used for experiment, the result of which is presented in Fig. 4. The expression of TLR4 was less consistent in the mouse model. These data are presented in supplemental figure S5. We sincerely thank you for the valuable suggestion, which has led to this important finding.
3- MAFB has been introduced as a prognostic marker in endometrial cancer. Could you please give some information about the role of MAFB in malignancies, including endometrial cancer?
Response: Thank you for your very important suggestion. The HPA data show that the higher the expression of MAFB, the lower is the survival rate of endometrial cancer. There are two possibilities as to how MAFB contributes to malignancy in such cancers. One is that MAFB is overexpressed and enhances progression through the cell cycle, as described for multiple myeloma in our response to your previous comment. The other possibility is that MAFB affects the tumor microenvironment. Recently, we have shown that MAFB is expressed in tumor associated macrophages (TAM) in lung cancer in mice and humans (Yadav et al., BBRC, 521, 590-595). TAM suppresses immunity, promotes angiogenesis and production of growth factors, and enhances tumorigenesis. Several studies, including the one by Steidl et al. (Steidl et al., N Engl J Med 362, 875-885, 2010), have shown that the higher the TAM level, the lower is the survival rate of patients. In addition, it is important to know whether MAFB is expressed in cancer cells or in TAMs, as the immunostaining results of MAFB in HPAs show that MAFB appears to be expressed outside the cancer cells. Furthermore, it was reported that the higher the number of TAMs in endometrial cancer, the lower is the survival rate (Kübler et al., Gynecologic Oncology 135, 176-183, 2014). This suggests that MAFB may serve as a tumor marker in endometrial cancer because it is expressed on TAMs rather than on tumor cells.
Results:
1- "It was reported that MAFB may be highly expressed in anti-inflammatory macrophages that suppress excessive inflammation and promote angiogenesis and cell proliferation [3,15]. Therefore, we hypothesized that MAFB contributes to anti-inflammatory macrophage function during wound healing." These phrases should be relocated to the discussion and not related to results. Moreover, I would like to suggest presenting the quantitative results in the 2.1 section with numbers and p-values.
Response: Thank you for your comments. We have removed the sentences from the Results section, as suggested. We have also rephrased the text explaining the data in section 2.1 and have added information about the sample numbers and p-values.
2- In figure 1A, please show the diameters with a scale to better correlate with Figure 1B. Figure 1D top does not have a scale bar and the unit of distances is not mentioned.
Response: Thank you for your suggestion. We have added a scale bar and mentioned the unit in Figure 1.
Discussion: Limitations of the study should be mentioned.
Response: Thank you for your important suggestion. We have added the limitations of the study in the last paragraph of the Discussion section.
Methods: How did you make sure that after cutting mice and making wounds, the other mice would not harm them?
Response: Thank you for your pertinent query. When we established the wound healing experiment, we discussed this point and conducted an experiment to see if there was any difference in wound healing between mice kept individually or in groups. No significant differences were observed between the two groups. Under the rearing conditions used by us, mice did not seem to be particularly interested in the wounds of other mice. In addition, we did not use male mice for the experiment because they might have fought once they were used for mating.
Reviewer 2 Report
The work is devoted to the study of the effect of the MAFB transcription factor on wound healing. The authors did a great job demonstrating that in MAFB-deficient mice, tissue healing is slower, and macrophage recruitment may depend on the expression of this transcription factor. The mechanism of its functioning is tied to the regulation of the expression of genes associated with the active process of wound healing. Technically, the experiments were performed qualitatively, which can be understood from clear and understandable visual materials. There are no complaints about the statistics either. The study is of interest from the point of view of the possibility of applying the results obtained in practice. Judging by the work of the authors, MAFB can indeed be considered as a therapeutic target, but this is still only the first step. Nevertheless, the work is interesting and deserves publication. The reviewer has no serious complaints about the quality of the research, and technical shortcomings can be corrected in the proofreading process, including checking the English language.
Author Response
Reviewer 2
The work is devoted to the study of the effect of the MAFB transcription factor on wound healing. The authors did a great job demonstrating that in MAFB-deficient mice, tissue healing is slower, and macrophage recruitment may depend on the expression of this transcription factor. The mechanism of its functioning is tied to the regulation of the expression of genes associated with the active process of wound healing. Technically, the experiments were performed qualitatively, which can be understood from clear and understandable visual materials. There are no complaints about the statistics either. The study is of interest from the point of view of the possibility of applying the results obtained in practice. Judging by the work of the authors, MAFB can indeed be considered as a therapeutic target, but this is still only the first step. Nevertheless, the work is interesting and deserves publication. The reviewer has no serious complaints about the quality of the research, and technical shortcomings can be corrected in the proofreading process, including checking the English language.
Response: Thank you for your generous comments. As you have mentioned, this study is a valuable first step toward MAFB becoming a therapeutic target in wound healing. We would like to continue our search for inducers of MAFB expression and would experiment with wound healing in disease conditions such as diabetes.